# Mechanistic-Based Classification of Endocytosis-Related Inhibitors: Does It Aid in Assigning Drugs against SARS-CoV-2?

**DOI:** 10.3390/v15051040

**Published:** 2023-04-23

**Authors:** Mohamed Hessien, Thoria Donia, Ashraf A. Tabll, Eiman Adly, Tawfeek H. Abdelhafez, Amany Attia, Samar Sami Alkafaas, Lucija Kuna, Marija Glasnovic, Vesna Cosic, Robert Smolic, Martina Smolic

**Affiliations:** 1Molecular Cell Biology Unit, Division of Biochemistry, Department of Chemistry, Faculty of Science, Tanta University, Tanta 31527, Egypt; 2National Research Centre, Microbial Biotechnology Department, Biotechnology Research Institute, Giza 12622, Egypt; 3Egypt Center for Research and Regenerative Medicine (ECRRM), Cairo 11517, Egypt; 4Department of Pharmacology and Biochemistry, Faculty of Dental Medicine and Health Osijek, University of J. J. Strossmayer Osijek, 31000 Osijek, Croatia; 5Department of Medicine, Family Medicine and History of Medicine, Faculty of Medicine Osijek, University of J. J. Strossmayer Osijek, 31000 Osijek, Croatia; 6Department of Paediatrics and Gynaecology with Obstetrics, Faculty of Dental Medicine and Health Osijek, University of J. J. Strossmayer Osijek, 31000 Osijek, Croatia; 7Department of Pharmacology, Faculty of Medicine Osijek, University of Osijek, 31000 Osijek, Croatia

**Keywords:** SARS-CoV-2, endocytosis, endocytic inhibitors, antiviral drugs, dynamin, clathrin

## Abstract

Severe Acute Respiratory Syndrome Coronavirus-2 (SARS-CoV-2) canonically utilizes clathrin-mediated endocytosis (CME) and several other endocytic mechanisms to invade airway epithelial cells. Endocytic inhibitors, particularly those targeting CME-related proteins, have been identified as promising antiviral drugs. Currently, these inhibitors are ambiguously classified as chemical, pharmaceutical, or natural inhibitors. However, their varying mechanisms may suggest a more realistic classification system. Herein, we present a new mechanistic-based classification of endocytosis inhibitors, in which they are segregated among four distinct classes including: (i) inhibitors that disrupt endocytosis-related protein–protein interactions, and assembly or dissociation of complexes; (ii) inhibitors of large dynamin GTPase and/or kinase/phosphatase activities associated with endocytosis; (iii) inhibitors that modulate the structure of subcellular components, especially the plasma membrane, and actin; and (iv) inhibitors that cause physiological or metabolic alterations in the endocytosis niche. Excluding antiviral drugs designed to halt SARS-CoV-2 replication, other drugs, either FDA-approved or suggested through basic research, could be systematically assigned to one of these classes. We observed that many anti-SARS-CoV-2 drugs could be included either in class III or IV as they interfere with the structural or physiological integrity of subcellular components, respectively. This perspective may contribute to our understanding of the relative efficacy of endocytosis-related inhibitors and support the optimization of their individual or combined antiviral potential against SARS-CoV-2. However, their selectivity, combined effects, and possible interactions with non-endocytic cellular targets need more clarification.

## 1. Introduction

SARS-CoV-2 and other viruses utilize different cellular endocytic pathways to invade their target cells [1]. Fine details of different endocytic mechanisms demonstrate how the plasma membrane (PM) is used to engulf viral particles, nutrients, or surface proteins to internalize them into the cell. These processes are well-orchestrated via many interacting cellular proteins that participate as receptors, scaffolding, adaptors, or modulatory proteins. This explains how it is possible to selectively inhibit clathrin-mediated endocytosis (CME) and other clathrin-independent mechanisms (Figure 1) at multiple stages to prevent viral invasion. In addition to RNA interference strategies (RNAi), the literature presents a long list of molecules that selectively inhibit different endocytic pathways that are predominantly used by SARS-CoV-2, such as CME and Caveolae-mediated endocytosis (CAE). As we will discuss later, the inhibition of endocytosis relies on the employment of small molecules (about 500 kDa or less) to interfere with the normal functions of key proteins or subcellular components that are directly involved in endocytosis. Clathrin and large dynamins, for example, are targeted by many inhibitors. Similarly, some clathrin and/or dynamin-independent endocytosis (CIE) inhibitors are shared with CME. Cholesterol, for example, is an integral constituent of the plasma membrane and lipid rafts; therefore, it is commonly targeted to inhibit both CME and some CIE mechanisms, including CAE. In addition, Filipin, a complex of at least four polyene macrolides [2], binds to cholesterol, disrupts the organization of the surrounding membrane, and inhibits CIE, whereas at higher concentrations, it also affects CME. In a similar manner, actin-depolymerizing compounds, such as Cytochalasin D and latrunculin, may represent key inhibitors for endocytic mechanisms [3]. Accordingly, most of the endocytic inhibitors are target-dependent, rather than endocytic pathway-dependent. In endocytosis-mediated viral entry into the target cells, these inhibitors may act in a viral type-independent manner.

The previously suggested classification of endocytic inhibitors relies on their chemical natures, whether they are chemical, pharmaceutical, or genetic inhibitors. They are also classified according to the endocytic pathway they are targeting [4]. Surprisingly, some FDA-approved drugs, such as Chlorpromazine (CPZ), chloroquine, and bisphosphonates, that are used in the treatment of psychotic disorders, malaria infection, and osteoporosis, respectively, and have more recently been suggested as SARS-CoV-2 drugs, are categorized as chemical inhibitors, rather than pharmaceutical. Furthermore, it is not clear why some inhibitors, such as the large dynamin inhibitors Dynasore and Dyngos, are categorized as pharmaceutical inhibitors. Moreover, there is no clear distinction between drugs that interrupt complex formation and those that inhibit the activity of endocytosis-related enzymes. This may raise the importance of suggesting a more justifiable classification that considers their mode of action. In this regard, the recent SARS-CoV-2 pandemic has spotlighted endocytosis and spurred tremendous efforts to present new, fast-acting, and reliable drugs, since some of the previously FDA-approved drugs were primarily considered endocytic inhibitors. Thus, this review suggests a new classification system of endocytic inhibitors and assigns many of the suggested SARS-CoV-2 therapeutic drugs to this classification. This perspective will enhance our understanding of their differential therapeutic effectiveness against SARS-CoV2 and other viruses.

## 2. Mechanistic-Based Classification of Endocytic Inhibitors

### 2.1. Concept of Classification

Careful inspection of the mechanisms of the currently identified endocytic mechanisms indicates that they utilize a group of cellular proteins and cellular components, such as the plasma membrane and endosome–lysosomal system. As CME is deeply concerned with receptor regulation, it has been extensively investigated. Accordingly, we utilized the main steps of CME as a guide to assign these inhibitors into four classes: First: inhibitors that interfere with protein–protein interactions, complex assembly or dissociation, which are involved in clathrin-coated pits (CCP), and vesicle formation. Second: inhibitors that prevent enzymatic activities associated with endocytosis. Third: inhibitors that function by disrupting the structure of subcellular components that are directly involved in endocytosis, such as the plasma membrane cholesterol and cytoskeletal actin. Fourth: inhibitors that alter the cellular metabolic and physiological conditions, leading to disturbed endocytosis (Figure 2).

### 2.2. Class I: Inhibitors of Complex Assembly and Dissociation

Protein–protein interaction (PPI) is a common phenomenon in molecular cell biology that results from the high specificity between domains of two or more proteins. The formed complex is stabilized by the conventional forces between amino acid residues. Due to their role in drug discovery, small molecules are employed as a therapeutic strategy to inhibit PPI (reviewed by Lu and others [5]). Following the same paradigm, many small molecules were presented to inhibit the constitutional stability and functions of endocytic complexes. Dynamin, clathrin, and AP2 adaptin are the most targeted endocytic proteins, and inhibitors targeting these complexes function in two ways: The first includes inhibitors that disturb the assembly of endocytic complexes involved in clathrin-coated pits (CCPs) formation to hinder cargo endocytosis. The second inhibits the uncoating process of the clathrin lattice, which halts the proper functioning of the endosomal–lysosomal system and leads to the inhibition of cargo degradation or recycling.

#### 2.2.1. Inhibitors of Complex Assembly

High-molecular-weight dynamins GTPases are a small family that includes Dyn1 which is expressed in neurons [6], Dyn2 which is expressed in all tissues [7], and Dyn3 which is expressed in the testis, lung, and heart [8]. Structurally, Dynamin comprises five conserved domains (Figure 3): a large N-terminal GTPase domain (G-domain); stalk; a Pleckstin homology (PH) domain (also, called foot) which binds phosphoinositides in the clathrin-coated vesicle (CCV) collar; a GTPase effector domain (GED); and a C-terminal proline-rich domain (PRD) [9]. In most eukaryotic cells, Dyn2 is involved, both structurally and enzymatically, in at least three endocytic pathways. Apart from its GTPase activity, Dyn2 is targeted by several compounds that inhibit its role in CCV fission. The earliest Dyn2 GTPase-independent inhibitors include ammonium salts, such as Myristyl trimethyl ammonium bromides (MiTMAB) and Octadecyltrimethylammonium bromide (OctMAB) [10]. Both compounds bind Dyn’s PH domain which is required for binding of Dyn2 to the plasma membrane [11]. This prevents the recruitment and oligomerization of Dyn2 around the vesicle neck. Notably, the PH domain is located in many proteins involved in intracellular signaling, for example AKT, and the cell cytoskeleton [12]. This may restrict the usability of PH domain-specific inhibitors as selective antiviral drugs.

Similarly, clathrin is targeted by several CME inhibitors. The term “Pitstop” was commercially coined to refer to at least four different compounds. (N-[5-[(4-Bromophenyl) methylene]-4,5-dihydro-4-oxo-2-thiazolyl]-1- naphthalene-sulfonamide) (Pitstop 2), for example, interacts with the N-terminal domain of the clathrin heavy chain, where it occupies the groove in which the clathrin-box motif (LfXfDE)-containing peptide of the adaptor protein binds [13]. Accordingly, the hindrance of the interaction between clathrin and adaptor proteins (e.g., amphiphysin) leads to the prevention of the formation of the clathrin lattice. In a similar context, Barbadin, was introduced as a potent CME inhibitor [14] as it inhibits β-arrestin/AP2 binding [15] through the inhibition of the interaction between the β2-adaptin ear domain of AP2 and β-arrestin. In this complex, β-arrestin fits a groove located in the β2-adaptin ear subdomain, where the FXXFXXXR motif from β-arrestin1 binds with the Q849, Y888, and Q902 residues of the β2-adaptin subunit [14]. Although Barbadin inhibits endocytosis of the agonist-activated receptors, such as β2-adrenergic (β2AR), V2-vasopressin (V2R), and angiotensin-II type-1 (AT1R) receptors, it does not demonstrate this effect on β-arrestin or AP2 independent receptor internalization. In addition, it does not inhibit the endocytosis of N-formyl peptide receptor *2* (FPR2) [16]. Moreover, in arginine vasopressin (AVP) stimulated breast cancer cells, we observed that Barbadin induced apoptosis, autophagy, and cell cycle arrest in the G0/G1 phase, indicating Barbadin’s side-talk with cell death-related targets [17].

#### 2.2.2. Inhibitors of Clathrin Lattice Uncoating

During the progression of CME, CCV fission leads to the formation of a cargo-loaded early endosome that delivers its contents to the cell membrane (recycling) or to the lysosome for degradation [18]. Before this sorting, it is essential to dissociate the clathrin lattice that covers the CCV. This process is mediated by Hsc70, and Auxilin [19,20]. Although chlorpromazine (CPZ) was previously FDA-approved as an antipsychotic drug, it can disrupt CME at multiple levels including the interaction between clathrin and AP2 [21], trapping the cargo inside the endosomes [22] and affecting Dyn2 GTPase activity [23]. Other studies showed that CPZ may induce AP2 depletion from the plasma membrane by blocking AP2 binding to an unidentified membrane-associated protein, which leads to the failure of clathrin import to the cell membrane [24].

### 2.3. Class II: Inhibitors of Enzyme Activities

#### 2.3.1. Large GTPases Inhibitors

Additionally, other family members are involved in other cellular aspects, including mitochondrial fission, the formation of new vesicles from the Golgi network, and the regulation of cytoskeleton dynamics. Therefore, mutations in Dyn genes are largely implicated in several human disorders As mentioned earlier, Dyn2 monomers are recruited to the neck of the nascent CCVs, forming a helical oligomer that catalyzes CCVs fission from the plasma membrane. In addition, after Dyn2 accumulation, the membrane is attached and cargo (or virus)-containing CCVs exhibit Dyn-mediated twisting, leading to vesicle fission [25]. As these scenarios are mediated by Dyn2’s GTPase activity, several compounds were introduced to inhibit Dyn’s GTPase activity. In 2006, 3-hydroxynaphthalene-2-carboxylic acid-(3,4-dihydroxybenzylidene)-hydrazide (Dynasore) (Figure 3) was presented as the first large dynamins GTPase inhibitor [26]. The advantages of Dynasore include its inability to affect small GTPases in addition to its revisable and [27] rapid effects as it inhibits CME in 2 min. Moreover, Dynasore reduces labile cholesterol in the plasma membrane and disrupts lipid raft organization in a Dyn-independent manner [28], which may enhance its inhibitory function. Later, Dynasore hydroxylated derivative (Dyngo) was developed and revealed more inhibitory potency and less cell cytotoxicity [29]. In addition, Dynole 34-2, was introduced as a potent large dynamin GTPase and ATPases inhibitor [30,31]. Similarly, Dynole 34-2 inhibits Dyn1 and Dyn2, where it targets their GTPase domain at the allosteric site. In the same context, some, less-commonly used Dynamin GTPase inhibitors were introduced, including Rhodadyns [32], 1,8-Naphthalimide derivatives (called Naphthaladyn) [23], Bisphosphonates drugs, used in the treatment of osteoporosis [33], Tyrphostins (BisT), Pthaladyns and Iminochromene [34,35] (Figure 4).

#### 2.3.2. Inhibitors of Kinases and Phosphatases

As mentioned earlier, some endocytic proteins are enzymatically modified to ensure their proper functions. In this regard, several enzymes, rather than GTPases, are involved in endocytosis. GPCR kinases, similar to GRK2, are expressed in many tissues and phosphorylate the intracellular domain of agonist-activated GPCR (Figure 5A), leading to the recognition of the viral–receptor complex by the N-terminal domain of β-arrestin [36]. The charged phosphate groups of the receptor induce the release of the C-terminal of the β-arrestin to facilitate its assembly with both clathrin and AP2 [37]. These events represent the initial steps of CCP formation. In the same context, Ark1/Prk1, a member of the Ser/Thr kinases family, is involved in phosphoryltion/dephosphorylation and ATPase activities [38]. Due to their role in controlling CME, they are targeted in anticancer/antiviral studies [39]. In addition, members of this family include human adaptor-associated kinase 1 (AAK1) that binds clathrin. Accordingly, active AAK1 phosphorylates Thr residue in the AP2-μ2-subunit, enhances the association with the cargo protein and plasma membrane PIP2, and creates the foundation of CCV [40]. Some AKK1 inhibitors, such as Liver kinase B1 (LKB1) and LP 935,509 (Figure 5B), are utilized as multi-kinase inhibitors [41] and have the potential to act as endocytic inhibitors. Similarly, AP2 dephosphorylation is one of the essential modifications required for CCVs uncoating. During this step, the previously phosphorylated AP2 µ-subunit is dephosphorylated, leading to its detachments from the cargo motif and clathrin [42].

In the same context, the heat-shock cognate (Hsc70) protein participates in many cellular processes, including ATP metabolism, protein folding, autophagy, and endocytosis. In CME, it acts as a key molecule in the dissociation of the clathrin lattice that coats CCV [43]. This dissociation involves Auxilin J-domain, which binds clathrin in the triskelions lattice, where the domain is exposed outward to facilitate the interaction with Hsc70 [44]. In addition, Hsc70 requires the QLMLT motif, located in the C-terminal of the clathrin heavy chain [45]. The lattice disassembly starts with bending at the location of insertion of “ankles”, followed by the interaction of Hsc70 with Auxilin near the C-terminus of the clathrin, where its strong interaction is mediated by its ATPase activity that facilitates the deformation of the clathrin lattice [46]. Thus, the functional association between Hsc70 and some of the CME-related proteins such as Clathrin and Auxillin [47] may predict the involvement of Hsc70 in endocytosis. Accordingly, some reports have suggested the antiviral potential of Hsc70 inhibitors such as tylophorine analogs [48] and the adenosine derivative compound VER-155008 [49]. In this regard, the direct participation of HSC70 in the SARS-CoV-2 infection is still unknown.

### 2.4. Class III: Inhibitors Targeting the Structure of Subcellular Components

In addition to the endocytic proteins, different endocytosis mechanisms involve various subcellular compartments, including the plasma membrane, cell cytoskeleton, endosomal–lysosomal system, and the Golgi apparatus. Additionally, there is a cross-talk between endocytosis and other cellular organelles, especially the mitochondria and the nucleus. As indicated below, several compounds were introduced to inhibit endocytosis by disrupting the structural integrity of the plasma membrane, lipid rafts, and actin cytoskeleton.

#### 2.4.1. Inhibitors Targeting the Plasma Membrane and Lipid Raft

Cholesterol is an integral component of the plasma membrane, where it modulates the fluidity, water penetration, and curvature of the lipid bilayers. It is also involved in membrane trafficking and protein sorting [50,51]. During endocytic pathways that are mediated by lipid rafts, such as CAE, membrane invaginations are initiated at the regions enriched with sphingolipids and cholesterol [52,53]. In addition, CLIC/GEEC endocytosis occurs in PIP2 or PIP3-rich sites and it largely relies on plasma membrane cholesterol [54]. Accordingly, the withdrawal of cholesterol from the plasma membrane was suggested as an effective strategy to inhibit viral-mediated endocytosis. Methyl-β-cyclodextrin (MBCD or βCD) [55] and Nystatin (Figure 6), for example, inhibit CME and effectively affect CAE and CLIC/GEEC via cholesterol extraction from the plasma membrane, where they form cholesterol–MBCD dimers [56,57]. Similarly, Filipin and 7-keto-cholesterol (Figure 6) bind to cholesterol and prevent the close packing of acyl chains, resulting in the inhibition of CLIC/GEEC [58,59,60]. In addition, 7-Ketocholesterol diminishes cholesterol synthesis, by inhibiting HMG-CoA reductase [61].

#### 2.4.2. Inhibitors of Actin Formation and Polymerization

Similar to withdrawal of the plasma membrane cholesterol, the actin assembly plays an essential role in endocytosis [62]. Many reports have investigated the notion of inhibiting clathrin-independent endocytosis by disrupting cytoskeletal actin formation or polymerization. Amiloride derivatives, such as 3-methylsulphonyl-4-piperidinobenzoyl guanidine hydrochloride, 5-(N-ethyl-N-isopropyl)-Amiloride (EIPA), Cytochalasin-D, and Latrunculin A and B (Figure 6B), affect actin formation or polymerization [63,64]. Cytochalasin-D, for example, binds to F-actin and prevents its polymerization, whereas Latrunculin-B is a potent actin-disrupting agent, sequesters monomeric G-actin, leading to massive disassembly of actin filament [65].

### 2.5. Class IV: Inhibitors That Alter Subcellular Physiological and Metabolic Homeostasis

Normal intracellular physiological conditions are necessary to ensure the proper functions of different cellular activities, including endocytosis. Hence, some factors may inhibit endocytosis through their ability to alter the normal metabolic and physiological conditions associated with endocytosis, including the subcellular compartment’s pH, ion concentration, osmolality across the plasma membrane, and several metabolic pathways such as the glycolytic pathways.

#### 2.5.1. Acidification Inhibitors

Normally, the cytosol acidity is maintained by the vacuolar H^+^-ATPases (V-ATPases), which are powered by ATP hydrolysis to transport protons from the cytosol to the interior of subcellular compartments such as lysosomes [66]. By default, lysosomes are acidic organelles with a pH of 4.5–5, whereas early and late endosomes are relatively less acidic at pH 6–6.5 and 5–6, respectively [67]. The relative acidity and pH gap of these organelles are tightly regulated to maintain the low acidic pH required for many cellular processes including the dissociation of the internalized liganded receptors, or endosomed viruses in the infected cells. Accordingly, many studies have suggested the role of acidification inhibitors in modulating different endocytic pathways. Endosidin9 (ES9), a mitochondrial uncoupler, for example, inhibits CME due to its protonophore activity that leads to cytoplasm acidification [68]. In addition, the polyether ionophoric antibiotic Monensin inhibits endocytosis and the SARS-CoV-2 infection by disrupting the proton gradient in the electron transport chain (ETC) [69,70]. More importantly, the inhibitory effect of chloroquine, one of the drugs suggested in the initial therapeutic protocols of the SARS-CoV-2 pandemic, relies on lysosomal acidification (discussed later) [71]. Other endosomal acidification inhibitors, including Bafilomycin-A1, ammonium chloride, and niclosamide (FDA-approved to treat tapeworm infestations), can strongly block clathrin and dynamin-independent endocytic mechanisms (e.g., the CLIC/GEEC (CG) pathway) and impede viral infection more than chloroquine [72].

#### 2.5.2. Inhibitors That Alter Ion Imbalance and Osmolality

The interrelation between hypokalemia and endocytosis was reported decades ago when early studies revealed that potassium depletion blocks CME and leads to the aggregation of clathrin in the cytoplasm, thus removing it from functioning in vesicle-coating [73] Some reports demonstrated that hypokalemia inhibited the endocytosis of the transferrin–transferrin receptor (Tf/TfR) complex and protected cells against poliovirus, but not rhinovirus type 2. In contrast, potassium depletion without hypotonic shock reduced the uptake of transferrin, and cells were susceptible to poliovirus infection [74]. These findings linked the hypokalemia/hypotonic condition with the inhibition of CME. Recently, some reports demonstrated the association of potassium concentration with SARS-CoV-2 infection [75], whereas other reports detected hypokalemia in only 41% of infected patients [76]. Similar to hypokalemia, hypertonic sucrose was reported as an inhibitor of endocytosis [77] as it inhibits the interaction between clathrin and AP2, leading to the inhibition of CCP formation [78]. Moreover, 2-deoxy-d-glucose/sodium azide inhibited the dispersion of clathrin lattices on the plasma membrane and trapped clathrin in microcages [79].

#### 2.5.3. Inhibitors of Cellular Metabolism

There is a reciprocal association between nutrient endocytosis and metabolism. The regular cellular uptake of nutrients (e.g., glucose, amino acids, etc.) and signaling molecules (e.g., hormones and growth factors) is mediated by membrane transporters, channels, or transmembrane receptors that show dynamic translocation between the plasma membrane and the cytosol. Iron homeostasis and cholesterol uptake, for example, are regulated through CME-mediated transferrin (TfR) and LDL (LDLR) receptors, respectively [80]. In addition, cellular uptake of glucose is regulated through the rate of glucose transporters endocytosis and exocytosis [81]. In this regard, hyperglycolysis, which is observed in SARS-CoV-2 patients, is considered a metabolic reprogramming of the glycolytic pathway, and it is associated with the severity of the disease. Additionally, in SARS-CoV-2 infected human monocytes, the virus stimulated glycolysis and upregulated the expression of glycolysis-related genes [82]. Accordingly, controlling hyperglycolysis was suggested as a therapeutic strategy, and some studies have suggested that glycolytic inhibitors are useful in decreasing ATP and NADH production [83]. Furthermore, glycolytic inhibitors, such as 2-deoxy-d glucose (2DG), attenuated SARS-CoV-2 multiplication in Vero E6 cells [84] and in human colorectal adenocarcinoma cells (human Caco-2 cells) [85]. The associated high production of reactive oxygen species (ROS) was targeted by higher doses of antioxidants [86].

## 3. Boundaries of SARS-CoV-2 Endocytosis and Susceptible Cells

### 3.1. Modes of SARS-CoV-2 Cell Entry

Viral entry into the target cells occurs either through endocytosis or membrane-mediated fusion. In the first (endocytosis-mediated entry), CME and CAE are the two most common endocytic pathways utilized by many viruses, including SARS-CoV-2. In this scenario, the entire SARS-CoV-2 viral particle invades the cell. After being engulfed in the endosome, the viral membrane fuses with the luminal face of the endosomal membrane, allowing for viral RNA transfer to the cytosol [87]. In addition to membrane receptors, some other cellular proteases may facilitate the SARS-CoV-2 entry process, including CD147, Neuropilin-1, Dipeptidyl peptidase-4 (DPP4), alanyl amino peptidase [88], the transmembrane protease, serine 2 (TMPRSS2) [89], and cathepsin L [90]. The initial steps of these events start with the elective binding of the viral spike (S)-protein to the Angiotensin-Converting (ACE2) receptor [91] and/or transferrin receptor [92]. This binding then triggers CME, similar to G-protein coupled receptors (GPCRs) internalization. In addition, some poorly characterized endocytic routes, such as clathrin and caveolae independent endocytic pathways, are involved in viral uptake [93,94]. Although macropinocytosis is not its key entry pathway, SARS-CoV-2 may activate the signaling pathways that trigger micropinocytosis-mediated infection [95]. Although, many investigators do not support viral entry by flotillin-dependent endocytosis, Glebov and his coworkers showed that SARS-CoV-2 may infect host cells using a flotillin-dependent mechanism [96]. Furthermore, viral entry may be established via the CLIC/GEEC pathway, which is pH sensitive but clathrin, dynamin and raft-independent [97], or via transcytosis, which internalizes microorganisms into the cell using membrane-bound carriers [98], to invade intestinal epithelial cells after viral binding to the ACE-2. Alternatively, SARS-CoV-2 entry takes place through viral fusion with the cell plasma membranes, and the viral RNA is delivered to the cytosol [99,100].

### 3.2. Cell Tropism of SARS-CoV-2

The respiratory system is a main target of SARS-CoV-2 infection, and some investigations have demonstrated that the ciliated and AT2 cells of the lung are the main targeted cells. In addition, basal, club, epithelial goblet, and ciliated cells of the trachea are infected with SARS-CoV-2 [101]. The susceptibility of these cells to viral infection is associated with the co-expression of the ACE2 receptor and the co-receptor transmembrane protease serine 2 (TMPRSS2) [102]. In the same context, accumulating evidence indicates the infectivity of the enterocytes of the small intestine [103], the distal tubular cells, and the collecting duct [104]. These observations imply that the cell tropism of SARS-CoV-2 for endocytosis greatly depends on receptor-mediated endocytosis.

## 4. Sorting Endocytic Proteins and Lipids as Anti-SARS-CoV-2 Targets

Although vaccination represents the gold standard strategy in viral infection management, a drug-based approach may offer an effective and fast alternative therapeutic tool. Antiviral drugs usually target the virus replication cycle or the host cell biology. Although it has some concerns, the latter approach may be advantageous due to the structural and physiological stability of the host cells compared with the constantly mutating viruses, variant emergence, and/or the development of drug resistance [105,106]. Over the past three years, many small molecules have been suggested to prevent SARS-CoV-2 infection or relieve the associated complications. Some of these drugs, such as chloroquine, were previously FDA-approved to treat several viral or nonviral-related human illnesses, and they were recalled in the clinical protocols of SARS-CoV-2 management. Considering the new classification of endocytic inhibitors, it is important to assign these drugs to different classes.

### 4.1. Potential SARS-CoV-2 Drugs Assigned to Class I

Although clathrin, dynamin 2, β-arrestin, and epstins are among the most important proteins in CME, the literature lists more than 50 endocytic accessory proteins involved in the initiation, progression, and release of cargo-containing endosomes [107]. Surprisingly, few inhibitors were observed to selectively inhibit the interaction between members of such complex machinery. These include Pistop 1, Pitstop 2, Barbadin, and Chlorpromazine (Table 1). Some reports, however, demonstrated that knocking out the clathrin heavy chain blocked CME and reduced SARS-CoV-2 infectivity [87,108]. In addition, Promethazine was suggested based on its ability to inhibit clathrin, and to reduce the symptoms and inflammation associated with SARS-CoV-2 infection [109]. Chlorpromazine, bolinaquinone (clathrin inhibitor), and Pitstop 2 are known to prevent the scission of the CCVs from the plasma membrane [110]. As expected, such compounds can block the entry of SARS-CoV-2 via the CME pathway as they demonstrated a partial decrease in the number of endosomed vesicles and minimized the severity of disease progression [111,112]. Unfortunately, some reports have shown that they were associated with harmful complications, including damage to the retina, photosensitivity, liver damage, seizures, headaches, stomach pain, and damage to the muscles or nerves [113]. Other compounds, such as Barbidin, demonstrated a potent and selective inhibition against β-arrestin and AP2 interaction during the initial steps of CCP formation; however, until now no reports have investigated Barbadin as an antiviral drug [114].

### 4.2. Potential SARS-CoV-2 Drugs Assigned to Class II

As Dynamin’s GTPase activity is involved in dynamin-dependent endocytic pathways, such as CME and CAE, several drugs were designed to inhibit Dynamin’s GTPase function. These include Dynasore, Dyngo™ (Hydroxylated Dynasore), Dynole 34-2, Rhodadyns, Naphthalimide derivatives (Naphthaladyn), Bisphosphonates and many other compounds (Table 1). Computational-based studies nominated Dynasore as a potent SARS-CoV-2 inhibitor [115]. Clinical trials have investigated the ability of fluvoxamine and Sertraline, to inhibit Dynamin GTPase activity in the treatment of SARS-CoV-2 [116,117]. Although GRK-mediated phosphorylation is considered a key event, none of the GRK selective inhibitors, such as mabuterol and 4-Amino-5-(bromomethyl)-2-methylpyrimi dine-dihydrobromide, were identified as potential CME-mediated SARS-CoV-2 inhibitors.

**Table 1 viruses-15-01040-t001:** Endocytic inhibitors including anti-SARS-CoV-2 drugs are assigned to different classes based upon their mechanism of action.

Class	Inhibitors
Class I	Pistop 1, Pitstop 2, Barbadin, Chlorpromazine*, Promethazine *, Ikarugamycin
Class II	Dynasore, Dyngo™ (Hydroxylated Dynasore), Dynole 34-2, Rhodadyns, Naphthalimide derivatives (Naphthaladyn), Bisphosphonates, Tyrphostins (BisT), Pthaladyns, Myristyl trimethyl ammonium bromides (MiTMAB), Octadecyltrimethyl-ammonium bromide (OctMAB), Iminochromene, 4-Amino-5-(bromomethyl)-2-methylpyrimidine-dihydrobromide, β2-adrenoreceptor agonists (Clenbuterol *, Fluvoxamine *, Brombuterol, Mabuterol, and Mapenterol), Liver kinase B1 (LKB1), LP-935509, Apoptozole, Rhodadyns, Paxlovid*
Class III	Methyl-β-cyclodextrin2 (MBCD, or βCD), Filipin, Nystatin *, 7-keto-cholesterol, cholesterol-25-hydroxylase, 3-methylsulphonyl-4-piperidinobenzoyl guanidine hydrochloride, 5-(N-ethyl-N-isopropyl)-Amiloride, Cytochalasin D and Latrunculin A and B, flubendazole *, terfenadine *, itraconazole *, vinblastine *, imipramine *
Class IV	Chloroquine *, Hydroxychloroquine *, Endosidin9, Bafilomycin-A1, ammonium chloride, Niclosamide *, hypertonic sucrose, 2-deoxy-D-glucose/sodium azide and Monensin *, Ouabain*, bufalin, Amiloride * (and Amiloride derivatives * EIPA and HOE-694).

(*): FDA-approved drugs.

### 4.3. Potential SARS-CoV-2 Drugs Assigned to Class III

Class III includes antiviral drugs that interfere with the structures of subcellular organelles deeply involved in endocytosis. In the plasma membrane, for example, the membrane rafts are enriched with both cholesterol and sphingolipid which act as portals for viral entry. Accordingly, several cholesterol-chelating agents were suggested as potential drugs for SARS-CoV-2 treatment, including methyl-β-cyclodextrin, phytosterols, and flavonoids, as they can block the entry of SARS-CoV-2 through CAE/lipid rafts [118]. Similarly, as the CLIC/GEEC (CG) pathway involves plasma membrane cholesterol, 7-keto-cholesterol inhibits CLIC/GEEC-mediated viral entry as it prevents the close packing of acyl chains and affects the structure of the cell membrane. In a similar manner, the antifungal drug Nystatin and vinblastine were utilized based on their role in cholesterol withdrawal [119]. Furthermore, inhibitors of plasma membrane ruffle formation, an early step in micropinocytosis, such as flubendazole, terfenadine, itraconazole, vinblastine and imipramine, block the entry of viruses into the host cells [71,120].

### 4.4. Potential SARS-CoV-2 Drugs Assigned to Class IV

As mentioned earlier, class IV inhibitors include drugs that inhibit endocytosis through the changes they induce in the physiological integrity of subcellular components associated with endocytosis. Disruption of the Na^+^/K^+^-ATPase (sodium pump) by cationic steroid inhibitors, such as ouabain and bufalin, led to the inhibition of CME and prevented SARS-CoV-2 entry [121]. In addition, pH-dependent ligand dissociation, which plays an integral role in the release of the endocytosed viral particles out of the endosome, is facilitated by low lysosomal pH [1]. This enables the virus to escape the endocytic pathway just before merging the viral-containing endosome with the lysosome. This scenario represents the strategy of basic lysosomotropic drugs in antiviral treatment. In this regard, several acidification inhibitors, such as chloroquine, BafilomycinA1, NH4Cl, and Niclosamide (FDA-approved to treat tapeworm infestations), facilitate lysosomal–endosomal merging, which leads to lysosomal-mediated viral degradation [122,123]. The antimalaria drug Chloroquine, in particular, was among the first few drugs investigated and even nominated as a drug to prevent SARS-CoV-2 infection [71]. Moreover, pH-dependent inhibitors affect the recruitment of clathrin and the associated adaptor proteins to the plasma membrane, and reduce the levels of PIP2 required for clathrin binding [123].

## 5. Summary and Future Perspectives

Eukaryotic cells utilize seven different endocytosis mechanisms for nutrient uptake, managing the relative abundance of membrane receptors and transporters, membrane remodeling, and neurotransmission. Normally, the cell simultaneously utilizes more than one endocytic pathway. Moreover, some cells adopt particular mechanisms rather than others to fulfill their basic functions. Intestinal enterocytes, for example, utilize pinocytosis to absorb fat droplets, whereas phagocytosis is predominantly used by antigen-presenting cells [124]. CME, however, is considered the major endocytosis mechanism due to its fundamental role in cell signaling, motility, cell–cell communication, and cell fate. Additionally, endocytosis is utilized by the pathogen to invade cells. Viruses, including SARS-CoV-2, invade cells via one or more of these endocytic pathways. There is consensus agreement about CME as the main mechanism utilized by SARS-CoV-2 to invade its target cells in addition to other clathrin and/or dynamin-independent endocytic pathways. Although vaccination is considered the gold standard protective strategy against viral infections, drug-based therapy is highly emphasized as a promising strategy to halt SARS-CoV-2 and other viruses’ endocytosis-mediated cell invasion.

In this review, we reclassified inhibitors of endocytosis, based on their mechanisms of action, into four main classes. In addition, many of the recently identified inhibitors that target SARS-CoV2 are assigned to the suggested classes. Class I includes the inhibitors that hinder the interaction between endocytic proteins required in the initial steps of endocytosis following the viral attachment to target cell receptors. Class II compounds inhibit the enzymatic activity associated with different endocytic pathways, particularly large dynamin GTPase activity. Class III compounds interfere with the structural integrity of the plasma membrane or actin polymerization. Class IV inhibitors modulate different cellular physiological conditions deeply involved in the endocytosis process.

The main characteristics of this classification are as follows: First, it is protein, lipid, pH, or ion channel-specific, rather than pathway specific. Second, it may accommodate any newly developed inhibitors, as long as they are targeting endocytosis-related events. Third, all small molecules that target endocytosis, function upstream of viral replication as they block either viral entry or its release from the endosome (pH-dependent). Fourth, this classification may aid in the better design of drug-based therapeutic protocols using a combination of inhibitors assigned to different classes to upgrade the limited successes of some inhibitors. Fifth, although we are focusing on SARS-CoV-2 infectivity, this classification could be viral type- and variant-independent.

This perspective does not underestimate drugs designed to halt viral replication; it provides a well-justified and aligned map of viral-entry blockers and opens the gate for more studies that explore their roles in viral replication and/or side-talk with cellular non-endocytic proteins. In addition, more investigations and computational studies may reveal the dual function of endocytosis inhibition and viral replication machinery.

## Figures and Tables

**Figure 1 viruses-15-01040-f001:**
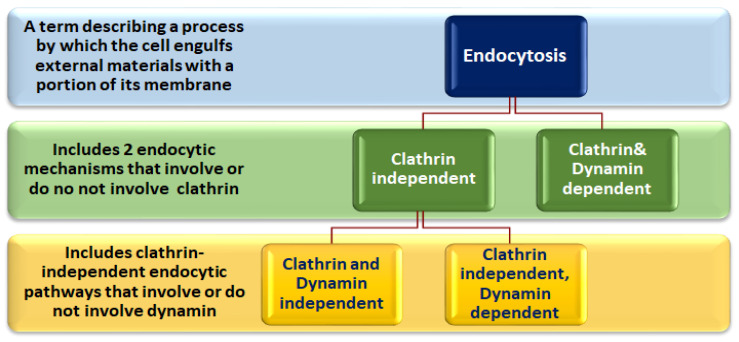
Classification of endocytic pathways based upon the involvement of clathrin and dynamin proteins. According to clathrin inclusion, endocytic pathways are classified into clathrin-dependent, such as CME, and clathrin-independent, whereas the clathrin-independent pathways are subdivided into clathrin and dynamin-independent (such as the CLIC/GEEC and Fillotin pathways) or clathrin-independent and dynamin-dependent, such as CAE. CME: Clathrin-mediated endocytosis; CLIC/GEEC: Clathrin-independent carriers/GPI anchored protein enriched early endosomal compartment; CAE: Caveolae-mediated endocytosis.

**Figure 2 viruses-15-01040-f002:**
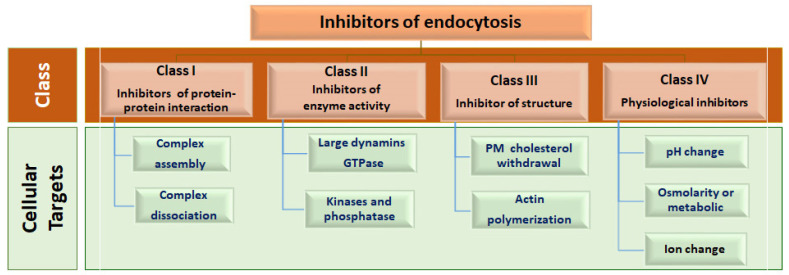
Mechanistic-based classification of endocytic inhibitors. Small molecule inhibitors targeting endocytosis are categorized into four main classes. The first class includes small molecules that interrupt the assembly or dissociation of the clathrin–adaptor complex. The second class includes compounds that suppress the GTPase, kinase, or phosphatase activities associated with endocytosis. The third and fourth classes include inhibitors that modulate the structural or physiological integrity, respectively, of subcellular compartments directly involved in endocytosis.

**Figure 3 viruses-15-01040-f003:**
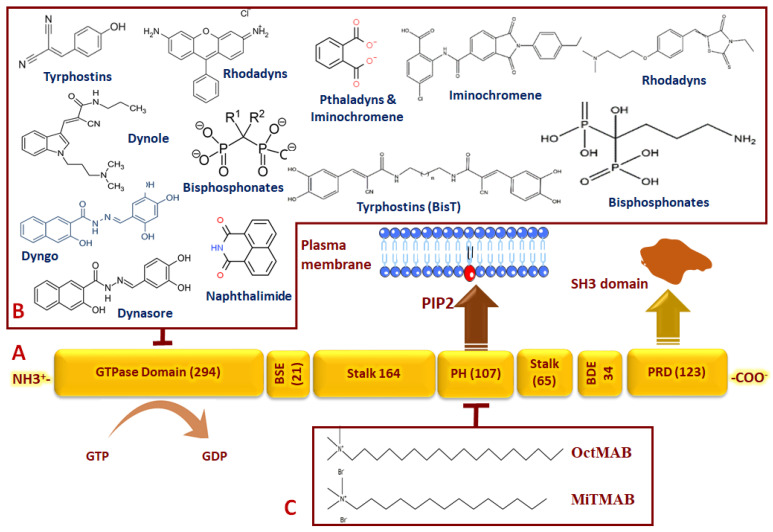
The basic organization of Dyn2 and its selective inhibitors. Dynamin 2 is a multimodular protein composed of five conserved domains including: a large N-terminal GTPase domain (G-domain), a middle domain, a PH domain (that is anchored with plasma membrane PIP 2), a GTPase effector domain (GED), and a C-terminal proline-rich domain (PRD) (**A**). The numbers of amino acid residues of each domain are shown. The sites of interactions of Dyn2 with the cell membrane and SH3-containing adaptor are indicated by arrows. GTPase-dependent or GTPase-independent inhibition is indicated by blunt arrows. Dyn2 GTPase activity is selectively inhibited by Dynasore and some other compounds (**B**), whereas both Octadecyltrimethylammonium bromide (OctMAB) and Myristyl trimethyl ammonium bromide (MitMAB) are GTPase-independent inhibitors, as they interact with the dynamin PH domain and prevent dynamin attachment to the plasma membrane (**C**).

**Figure 4 viruses-15-01040-f004:**
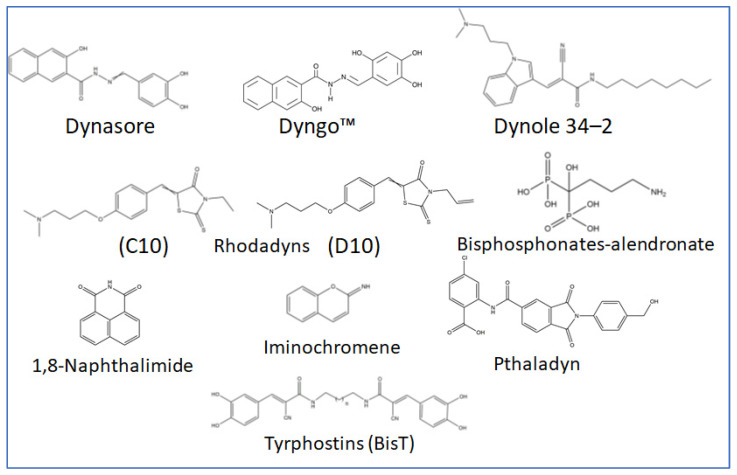
Chemical structures of Dynamin GTPase inhibitors. Dynamin GTPase activity is inhibited by a variety of small molecules including: Dynasore, Dyngo, Dynole 34-2, Rhodadyns, Bisphosphonates- alendronate, Tyrphostins, Iminochromene, 1,8-Naphthalimide and Pthaladyn.

**Figure 5 viruses-15-01040-f005:**
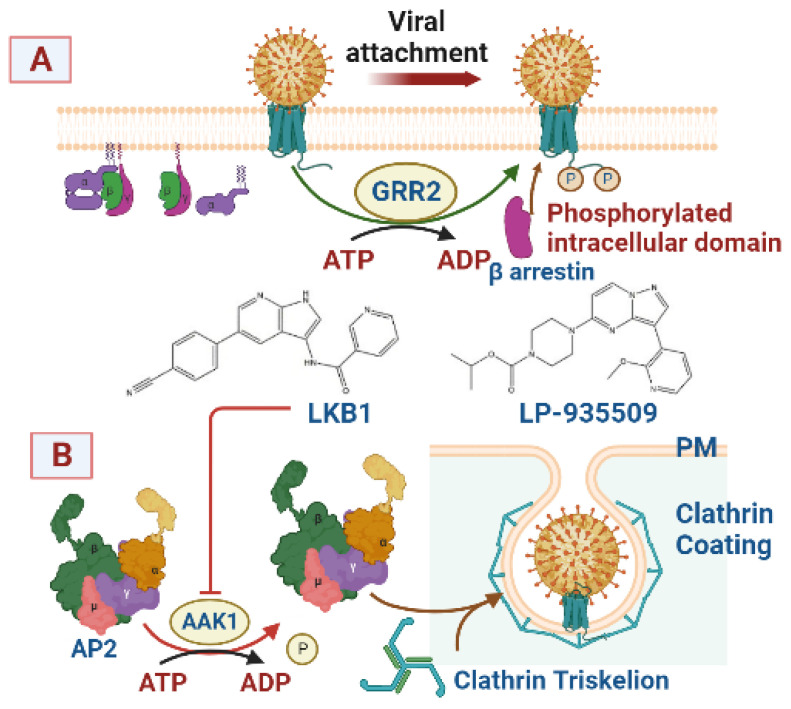
Kinase- and phosphatase-mediated modulation of endocytosis-related proteins. (**A**) G protein-coupled receptor kinases (GRKs) phosphorylate the intracellular domain of the ligand-activated receptor, leading to β-arrestin-mediated cargo recognition. (**B**) Human adaptor-associated kinase 1 (AAK1) phosphorylates the AP2 μ2-subunit and triggers the formation of clathrin-coated pits (CCP). First, clathrin binds to AAK1 leading to its activation and consequently phosphorylation of AP2 µ2 at the Thr residue and its binding to cargo protein and PIP2 in the plasma membrane (PM). (**C**) Chemical structure of LKB1 and LP-935509 that selectively inhibit adaptor-associated kinase 1 (AAK1).

**Figure 6 viruses-15-01040-f006:**
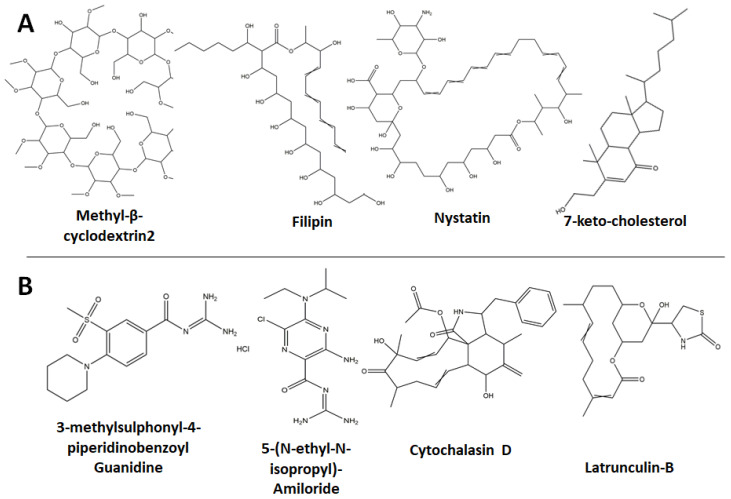
Inhibitors that disrupt the integrity of subcellular structures. Plasma membrane cholesterol withdrawal compounds (**A**) include Methyl-β-cyclodextrin2, Filipin, 7-keto-cholesterol, and Nystatin. Other compounds such as 3-methylsulphonyl-4-piperidinobenzoyl Guanidine, 5-(N-ethyl-N-isopropyl)-Amiloride (EIPA), Cytochalasin D, and Latrunculin-B (bottom panel) restrict actin formation or polymerization (**B**).

## Data Availability

Not applicable.

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
