# Peer review of "Mechanistic-Based Classification of Endocytosis-Related Inhibitors: Does It Aid in Assigning Drugs against SARS-CoV-2?"

_viruses, 2023, doi:10.3390/v15051040_

Round 1

Reviewer 1 Report

Based on their mechanism of action, this review reclassified endocytosis inhibitors into four main classes. And, the currently identified inhibitors that target SARS-CoV-2 are assigned to the suggested classes. This classification may aid in better designing of drug-based therapeutic protocols using a combination of inhibitors assigned to different classes to upgrade the limited successes of some inhibitors. The manuscript was basically well written, and demonstrates an mechanism-based classification review in the field of endocytosis-related inhibitors. This manuscript could be considered for publishing in Viruses, after some minor revisions.

1. All the structural formulas of the compounds in the manuscript need to be redrawn in a consistent format using Chemdraw, especially Figures 4 and 5. Currently, there are many structural formulas in screenshots from other articles;

2. The structure of Chromamine is not showed and the structural formula of Rhodadyns is incomplete in figure 5;

3. Many figures are not clear, especially Figure 6;

4. The cellular activity data of the SRAS-CoV-2 inhibitors mentioned in the manuscript need to be provided.

5. There are too many errors in Table 2, such as SRAS-CoV2, SRAS-Cov-2, and the last few references in the table.

Reviewer 2 Report

The manuscript entitled: “Mechanist-based classification of endocytosis-related inhibitors: Does it aid in assigning drugs against SARS-CoV-2?”  by Hessien et al. concerns an interesting topic, being the endocytosis an essential step for the viral replication of many viruses, including SARS-CoV-2 and therefore it represents a potential target of antiviral molecules. The manuscript is potentially suitable for the publication on Viruses with major revisions listed below. Furthermore, the english language needs profound revisions. I strongly suggest to submit the manuscript to a language and grammar check by a native English speaker.

Figure 1: remove PPI from the figure, as it is the same as the title of its category (class I).

Figure 3: this figure is redundant. Figure 4 clearly and concisely depicts the same concepts. I suggest to remove this figure.

Figure 4: the legend is wrong.Dynamin-2 GTPase activity is selec-tively inhibited by Dynasore (A)”: change A into B.

Figure 7: the figure is not clear. The authors should clearly define the legend, as the reader cannot distinguish between the two categories of molecules depicted (i.e., plasma membrane cholesterol withdrawal compounds (indicated in the legend as top panel) and actin inhibitors (indicated in the legend as bottom panel).

Table 2, “Endocytic inhibitors assigned to different classes based upon their mechanism of action.”: this table should be removed, as it is not unclear and reductive. It’s unclear why only some viruses, other than SARS-CoV-2, are cited, what is the role of receptors in the endocytosis, and what is the overall meaning of the table in this context.

I suggest to move the section 2.3 class II before the section 2.2 class I, as Dynamin is introduced in more details in section 2.3 rather than section 2.2.

Page 4, section 2.2.1: “Pitstopt-similar compounds demonstrated different physicochemical characteristics including, structures and membrane diffusion potentials.” Are there any references to this statement? Otherwise, I would suggest to remove the statement as it is not highly relevant to the subject.

Page 6, section 2.3.1: reduce the paragraph of the description of Dynamin. The sentence “Therefore, mutations in Dyn genes are largely implicated in several human disorders [27].” is not relevant.

Page 7: sections 2.3.2.1 and 2.3.2.2 and 2.3.2.3 can be combined together and reduced for conciseness.

Page 7: “Although GRK-mediated phosphorylation is considered a key event, none of GRK selective inhibitors (like mabuterol and 4-Amino-5-(bromomethyl)-2-methylpyrimidine-dihydrobromide) were nominated as potential CME-mediated SARS-CoV-2 inhibitors.” This sentence should be moved in the appropriate section on SARS-CoV-2 (section 4.2).

Page 10, section 2.5: remove the bold font.

Page 10, sections 2.5.1 and 2.5.2: remove the references to viruses, and leave only the description of the general endocytosis processes. Move the references to viruses and pH acidification/ion imbalance in the appropriate sections on SARS-CoV-2 (sections 3, 4.3 and 4.4).

Page 11: sections 2.5.2.1 and 2.5.2.2 can be combined together for conciseness.

Page 11, section 2.5.3: correct the title of the section, changing “Inhibition” with “Inhibitors”.

Page 11, section 3: “Viral entry into the target cells occurs either through the delivery of their genome into the cytosol or through viral endocytosis.” This sentence is unclear: did the authors mean through fusion or endocytosis? Please clarify in the text.

Page 11, section 3: Given the main virological interest of this journal, authors should expand this paragraph including more details on the mechanisms employed by SARS-CoV-2 in the endocytotic entry of host cells. Moreover, authors should comment on the fact that fusion is also reported as an entry mechanism of SARS-CoV-2 and how this could affect the efficacy of endocytosis inhibitors. Finally, different cellular and in vivo models are used to study the viral endocytosis inhibitors: authors should comment on how the different model could affect the efficacy of such inhibitors.

Page 14, section 4.4: authors should further report Chloroquine and other pH acidification inhibitors in this paragraph, as this was one of the first and main classes of endocytosis inhibitors studied for SARS-CoV-2.

Page 14, section 5: “Although vaccination is considered the gold standard treatment of viral infections,” this sentence should be removed as it is improper to state that vaccination is a treatment for viral infections.

Reviewer 3 Report

Please see the uploaded file.

Round 2

Reviewer 2 Report

The last version of the article requires further editing of English language.